# Deconstructing Data Reconstruction: Multiclass, Weight Decay and General Losses

**Gon Buzaglo**[*][1]    **Niv Haim**[*][1]    **Gilad Yehudai**[1]    **Gal Vardi**[2]

**Yakir Oz**[1]    **Yaniv Nikankin**[1]    **Michal Irani**[1]

[1]Weizmann Institute of Science, Rehovot, Israel
[2]TTI-Chicago and the Hebrew University of Jerusalem

## Abstract

Memorization of training data is an active research area, yet our understanding of the inner workings of neural networks is still in its infancy. Recently, Haim et al. [2022] proposed a scheme to reconstruct training samples from multilayer perceptron binary classifiers, effectively demonstrating that a large portion of training samples are encoded in the parameters of such networks. In this work, we extend their findings in several directions, including reconstruction from multiclass and convolutional neural networks. We derive a more general reconstruction scheme which is applicable to a wider range of loss functions such as regression losses. Moreover, we study the various factors that contribute to networks' susceptibility to such reconstruction schemes. Intriguingly, we observe that using weight decay during training increases reconstructability both in terms of quantity and quality. Additionally, we examine the influence of the number of neurons relative to the number of training samples on the reconstructability.
Code: https://github.com/gonbuzaglo/decoreco

## 1 Introduction

Neural networks are known to memorize training data despite their ability to generalize well to unseen test data [Zhang et al., 2021, Feldman, 2020]. This phenomenon was observed in both supervised settings [Haim et al., 2022, Balle et al., 2022, Loo et al., 2023] and in generative models [Carlini et al., 2019, 2021, 2023]. These works shed an interesting light on generalization, memorization and explainability of neural networks, while also posing a potential privacy risk.

Current reconstruction schemes from trained neural networks are still very limited and often rely on unrealistic assumptions, or operate within restricted settings. For instance, Balle et al. [2022] propose a reconstruction scheme based on the assumption of having complete knowledge of the training set, except for a single sample. Loo et al. [2023] suggest a scheme which operates under the NTK regime [Jacot et al., 2018], and assumes knowledge of the full set of parameters at initialization. Reconstruction schemes for unsupervised settings are specifically tailored for generative models and are not applicable for classifiers or other supervised tasks.

Recently, Haim et al. [2022] proposed a reconstruction scheme from feed-forward neural networks under logistic or exponential loss for binary classification tasks. Their scheme requires only knowledge of the trained parameters, and relies on theoretical results about the implicit bias of neural networks towards solutions of the maximum margin problem [Lyu and Li, 2019, Ji and Telgarsky, 2020]. Namely, neural networks are biased toward KKT points of the max-margin problem (see Theorem 3.1). By utilizing the set of conditions that KKT points satisfy, they devise a novel loss function that allows for reconstruction of actual training samples. They demonstrate reconstruction from models trained on common image datasets (CIFAR10 [Krizhevsky et al., 2009] and MNIST [LeCun et al., 2010]).

---

[*]Equal Contribution

37th Conference on Neural Information Processing Systems (NeurIPS 2023).

In this work, we expand the scope of neural networks for which we have evidence of successful sample memorization, by demonstrating sample reconstruction. Our contributions are as follows:

- We extend the reconstruction scheme of Haim et al. [2022] to a multiclass setting (Fig. 1). This extension utilizes the implicit bias result from Lyu and Li [2019] to multiclass training. We analyse the effects of the number of classes on reconstructability, and show that models become more susceptible to sample reconstruction as the number of classes increases.

- We devise a reconstruction scheme that applies for general loss functions, assuming that the model is trained with weight decay. We demonstrate reconstruction from models trained on regression losses.

- We investigate the effects of weight decay and show that for certain values, weight decay increases the vulnerability of models to sample reconstruction. Specifically, it allows us to reconstruct training samples from a convolutional network, while Haim et al. [2022] only handled MLPs.

- We analyse the intricate relation between the number of samples and the number of parameters in the trained model, and their effect on reconstrctability. We also demonstrate successful reconstruction from a model trained on 5,000 samples, surpassing previous results that focused on models trained on up to 1,000 samples.

## 2 Related Works

**Memorization and Samples Reconstruction.** There is no consensus on the definition of the term "memorization" and different works study this from different perspectives. In ML theory, memorization usually refers to label (or, model's output) memorization [Zhang et al., 2016, Arpit et al., 2017, Feldman, 2020, Feldman and Zhang, 2020, Brown et al., 2021], namely, fitting the training set. Memorization in the *input* domain is harder to show, because in order to demonstrate its occurrence one has to reconstruct samples from the model. Balle et al. [2022] demonstrated reconstruction of one training sample, assuming knowledge of all other training samples and Haim et al. [2022] demonstrated reconstruction of a substantial portion of training samples from a neural network classifier. Loo et al. [2023] extend their work to networks trained under the NTK regime [Jacot et al., 2018] and explore the relationship to dataset distillation. Several works have also studied memorization and samples reconstruction in generative models like autoencoders [Erhan et al., 2009, Radhakrishnan et al., 2018], large language models [Carlini et al., 2021, 2019, 2022] and diffusion-based image generators [Carlini et al., 2023, Somepalli et al., 2022, Gandikota et al., 2023, Kumari et al., 2023].

**Inverting Classifiers.** Optimizing a model's input as to minimize a class score is the common approach for neural network visualization [Mahendran and Vedaldi, 2015]. It usually involves using input regularization [Mordvintsev et al., 2015, Ghiasi et al., 2022], GAN prior [Nguyen et al., 2016, 2017] or knowledge of batch statistics [Yin et al., 2020]. Fredrikson et al. [2015], Yang et al. [2019] showed reconstruction of training samples using similar approach, however these methods are limited to classifiers trained with only a few samples per class. Reconstruction from a federated-learning setup [Zhu et al., 2019, He et al., 2019, Hitaj et al., 2017, Geiping et al., 2020, Yin et al., 2021, Huang et al., 2021, Wen et al., 2022] involve attacks that assume knowledge of training samples' gradients (see also Wang et al. [2023] for a theoretical analysis). In this work we do not assume any knowledge on the training data and do not use any priors other than assuming bounded inputs.

## 3 Preliminaries

In this section, we provide an overview of the fundamental concepts and techniques required to understand the remainder of the paper, focusing on the fundamentals laid by Haim et al. [2022] for reconstructing training samples from trained neural networks.

**Theoretical Framework.** Haim et al. [2022] builds on the theory of implicit bias of gradient descent. Neural networks are commonly trained using gradient methods, and when large enough, they are expected to fit the training data well. However, it is empirically known that these models converge to solutions that also generalize well to unseen data, despite the risk of overfitting. Several

works pointed to this "*implicit bias*" of gradient methods as a possible explanation. Soudry et al. [2018] showed that linear classifiers trained with gradient descent on the logistic loss converge to the same solution as that of a hard-SVM, meaning that they maximize the margins. This result was later extended to non-linear and homogeneous neural networks by Lyu and Li [2019], Ji and Telgarsky [2020]:

**Theorem 3.1 (Paraphrased from Lyu and Li [2019], Ji and Telgarsky [2020])** *Let $\Phi(\boldsymbol{\theta}; \cdot)$ be a homogeneous* [2] *ReLU neural network. Consider minimizing the logistic loss over a binary classification dataset $\{(\mathbf{x}_i, y_i)\}_{i=1}^{n}$ using gradient flow. Assume that there exists time $t_0$ where the network classifies all the samples correctly. Then, gradient flow converges in direction to a first order stationary point (KKT point) of the following maximum-margin problem:*

$$\min_{\boldsymbol{\theta}} \frac{1}{2} \|\boldsymbol{\theta}\|^2 \quad s.t. \quad \forall i \in [n] \ \ y_i \Phi(\boldsymbol{\theta}; \mathbf{x}_i) \geq 1 \ . \tag{1}$$

A KKT point of Eq. (1) is characterized by the following set of conditions:

$$\forall j \in [p], \ \ \boldsymbol{\theta}_j - \sum_{i=1}^{n} \lambda_i \nabla_{\boldsymbol{\theta}_j} [y_i \Phi(\boldsymbol{\theta}; \mathbf{x}_i)] = 0 \qquad \text{(stationarity)} \tag{2}$$

$$\forall i \in [n], \ \ y_i \Phi(\boldsymbol{\theta}; \mathbf{x}_i) \geq 1 \qquad \text{(primal feasibility)} \tag{3}$$

$$\forall i \in [n], \ \ \lambda_i \geq 0 \qquad \text{(dual feasibility)} \tag{4}$$

$$\forall i \in [n], \ \ \lambda_i = 0 \text{ if } y_i \Phi(\boldsymbol{\theta}; \mathbf{x}_i) \neq 1 \qquad \text{(complementary slackness)} \tag{5}$$

**Reconstruction Algorithm.** Haim et al. [2022] demonstrated reconstructing samples from the training set of such classifiers by devising a reconstruction loss. Given a trained classifier $\Phi(\mathbf{x}; \boldsymbol{\theta})$, they initialize a set of $\{(\mathbf{x}_i, y_i)\}_{i=1}^{m}$ and $\{\lambda_i\}_{i=1}^{m}$, and optimize $\mathbf{x}_i, \lambda_i$ to minimize the following loss function:

$$L = \underbrace{\|\boldsymbol{\theta} - \sum_{i=1}^{m} \lambda_i \nabla_{\boldsymbol{\theta}_j} [y_i \Phi(\boldsymbol{\theta}; \mathbf{x}_i)] \|}_{L_{\text{stationary}}} + \underbrace{\sum_{i=1}^{m} \max\left\{-\lambda, -\lambda_{\min}\right\}}_{L_\lambda} + L_{\text{prior}} \tag{6}$$

Where $L_{\text{prior}}$ is simply bounding each pixel value at $[-1, 1]$ [3]. The number of training samples $n$ is unknown. However, setting $m > 2n$, where $\{y_i\}$ are set in a balanced manner allows reconstructing samples with any label distribution. The $\mathbf{x}_i$'s are initialized from the Gaussian distribution $\mathcal{N}(0, \sigma_x^2 \mathbb{I})$, and $\lambda_{\min}, \sigma_x$ are hyperparameters. We note that the homogeneity condition from Theorem 3.1 is not necessarily a practical limitation of this reconstruction scheme, as already in Haim et al. [2022] they show reconstructions from a non-homogeneous network.

**Analysing and Summarizing Results.** The optimization in Eq. (6) is executed $k$ times (for different hyperparameters) and results in $km$ outputs ($\{\hat{\mathbf{x}}_i\}_{i=1}^{km}$) that we term *candidates*, as they are candidates to be reconstructed training samples. To quantify the success of the reconstruction process, each training sample is matched with its nearest-neighbour from the $km$ candidates. The "quality" of reconstruction is then measured using SSIM Wang et al. [2004] (see full details in Appendix A.1).

An important corollary of the set of KKT conditions Eqs. (2) to (5) is that the parameters of the trained model only depend on gradients of samples that are closest to the decision boundary, the so-called "margin-samples" (see end of section 3.2 in Haim et al. [2022]). Therefore, a good visual summary for analysing reconstruction from a trained model is by plotting the reconstruction quality (SSIM) against the distance from the decision boundary ($|\Phi(\mathbf{x}_i)|$). We also utilize such visualizations.

**Assessing the Quality of Reconstructed Samples.** Determining whether a candidate is a correct match for some training sample is as hard as finding a good image similarity metric. No synthetic metric such as SSIM or L2 norm can perfectly align with human perception. Perceptual similarity

---

[2] A classifier $\Phi$ is homogeneous w.r.t to $\boldsymbol{\theta}$ if there exists $L \in \mathbb{R}$ s.t. $\forall c \in \mathbb{R}, \mathbf{x} : \phi(\mathbf{x}; c\boldsymbol{\theta}) = c^L \phi(\mathbf{x}; \boldsymbol{\theta})$

[3] Formally: $L_{\text{prior}} = \sum_{i=1}^{m} \sum_{k=1}^{d} \max\{\max\{\mathbf{x}_{i,k} - 1, 0\}, \max\{-\mathbf{x}_{i,k} - 1, 0\}\}$.

metrics (e.g., LPIPS [Zhang et al., 2018]) build on top of pre-trained classifiers trained on Imagenet [Deng et al., 2009], and are not effective for the image resolution in this work (up to 32x32 pixels). We have observed heuristically that candidates with SSIM score higher than about 0.4 are indeed visually similar to their nearest neighbor training sample. Hence, in this work we say that a certain candidate is a *"good reconstruction"* if its SSIM score with its nearest neighbor is at least 0.4. Also see discussion in Appendix A.2.

# 4 Reconstructing Data from Multi-Class Classifiers

We demonstrate that training set reconstruction can be extended to multi-class classification tasks.

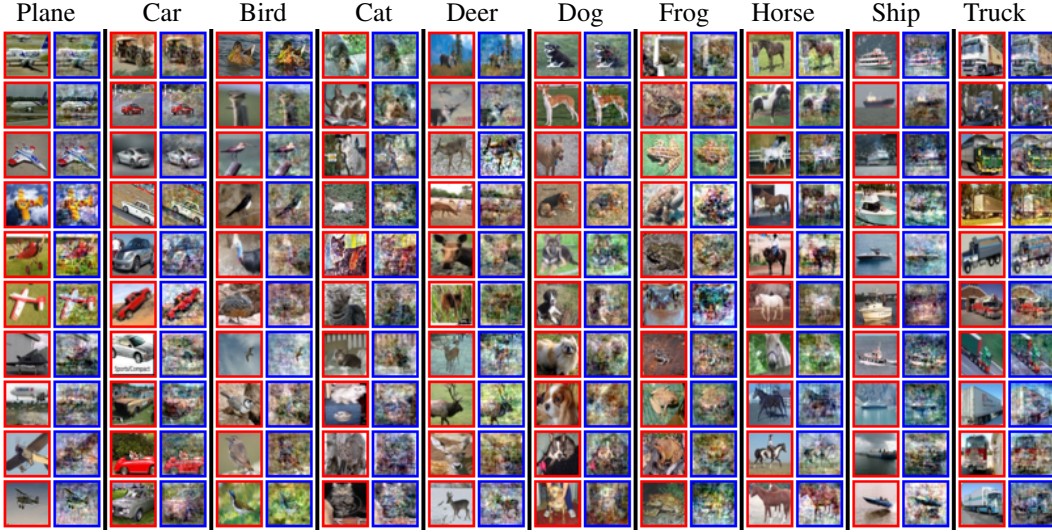

Figure 1: Reconstructed training samples from a multi-class MLP classifier that was trained on 500 CIFAR10 images. Each column corresponds to one class and shows the 10 training samples (*red*) that were best reconstructed from this class, along with their reconstructed result (*blue*).

## 4.1 Theory

The extension of the implicit bias of homogeneous neural networks to the multi-class settings is discussed in Lyu and Li [2019] (Appendix G): Let $S = \{(\mathbf{x}_i, y_i)\}_{i=1}^n \subseteq \mathbb{R}^d \times [C]$ be a multi-class classification training set where $C \in \mathbb{N}$ is any number of classes, and $[C] = \{1, \dots, C\}$. Let $\Phi(\boldsymbol{\theta}; \cdot) : \mathbb{R}^d \to \mathbb{R}^C$ be a homogeneous neural network parameterized by $\boldsymbol{\theta} \in \mathbb{R}^p$. We denote the $j$-th output of $\Phi$ on an input $\mathbf{x}$ as $\Phi_j(\boldsymbol{\theta}; \mathbf{x}) \in \mathbb{R}$. Consider minimizing the standard cross-entropy loss and assume that after some number of iterations the model correctly classifies all the training examples. Then, gradient flow will converge to a KKT point of the following maximum-margin problem:

$$\min_{\boldsymbol{\theta}} \frac{1}{2} \|\boldsymbol{\theta}\|^2 \quad \text{s.t.} \quad \Phi_{y_i}(\boldsymbol{\theta}; \mathbf{x}_i) - \Phi_j(\boldsymbol{\theta}; \mathbf{x}_i) \geq 1 \quad \forall i \in [n], \forall j \in [C] \setminus \{y_i\} \quad . \tag{7}$$

A KKT point of the above optimization problem is characterized by the following set of conditions:

$$\boldsymbol{\theta} - \sum_{i=1}^n \sum_{j \neq y_i}^c \lambda_{i,j} \nabla_{\boldsymbol{\theta}} (\Phi_{y_i}(\boldsymbol{\theta}; \mathbf{x}_i) - \Phi_j(\boldsymbol{\theta}; \mathbf{x}_i)) = \mathbf{0} \tag{8}$$

$$\forall i \in [n], \forall j \in [C] \setminus \{y_i\} : \quad \Phi_{y_i}(\boldsymbol{\theta}; \mathbf{x}_i) - \Phi_j(\boldsymbol{\theta}; \mathbf{x}_i) \geq 1 \tag{9}$$

$$\forall i \in [n], \forall j \in [C] \setminus \{y_i\} : \quad \lambda_{i,j} \geq 0 \tag{10}$$

$$\forall i \in [n], \forall j \in [C] \setminus \{y_i\} : \quad \lambda_{i,j} = 0 \text{ if } \Phi_{y_i}(\boldsymbol{\theta}; \mathbf{x}_i) - \Phi_j(\boldsymbol{\theta}; \mathbf{x}_i) \neq 1 \tag{11}$$

A straightforward extension of a reconstruction loss for a multi-class model that converged to the conditions above would be to minimize the norm of the left-hand-side (LHS) of condition Eq. (8)

(namely, optimize over $\{\mathbf{x}_i\}_{i=1}^m$ and $\{\lambda_{i,j}\}_{i\in[n],j\in[C]\setminus y_i}$ where $m$ is a hyperparameter). However, this straightforward extension failed to successfully reconstruct samples. We therefore propose the following equivalent formulation.

Note that from Eqs. (9) and (11), most $\lambda_{i,j}$ zero out: the distance of a sample $\mathbf{x}_i$ to its nearest decision boundary, $\Phi_{y_i} - \max_{j\neq y_i} \Phi_j$, is usually achieved for a single class $j$ and therefore (from Eq. (11)) in this case at most one $\lambda_{i,j}$ will be non-zero. For some samples $\mathbf{x}_i$ it is also possible that all $\lambda_{i,j}$ will vanish. Following this observation, we define the following loss that only considers the distance from the decision boundary:

$$L_{\text{multiclass}}(\mathbf{x}_1, ..., \mathbf{x}_m, \lambda_1, ..., \lambda_m) = \left\| \boldsymbol{\theta} - \sum_{i=1}^m \lambda_i \, \nabla_{\boldsymbol{\theta}}[\Phi_{y_i}(\mathbf{x}_i; \boldsymbol{\theta}) - \max_{j\neq y_i} \Phi_j(\mathbf{x}_i; \boldsymbol{\theta})] \right\|_2^2 \qquad (12)$$

Eq. (12) implicitly includes Eq. (11) into the summation in Eq. (8), thereby significantly reducing the number of summands and simplifying the overall optimization problem.

While the straightforward extension failed to successfully reconstruct samples, solving Eq. (12) enabled reconstruction from multiclass models (see Fig. 1 and results below). We attribute this success to the large reduction in the number of optimization variables, which simplifies the overall optimization problem ($n$ variables in Eq. (12) compared to $C \cdot n$ variables in the straightforward case, which is significant for large number of classes $C$).

We also use the same $L_\lambda$ and $L_{\text{prior}}$ as in Eq. (6), and set $\{y_i\}$ in a balanced manner (uniformly on all classes). While setting $m = C \cdot n$ allows reconstructing any label distribution, in our experiments we focus on models trained on balanced training sets, and use $m = 2n$ which works sufficiently well. An intuitive way to understand the extension of the binary reconstruction loss Eq. (6) to multi-class reconstruction Eq. (12) is that the only difference is the definition of the *distance to nearest boundary*, which is the term inside the square brackets in both equations.

## 4.2 Results

We compare between reconstruction from binary classifiers, as studied in Haim et al. [2022], and reconstruction from multi-class classifiers by using the novel loss function Eq. (12). We conduct the following experiment: we train an MLP classifier with architecture $D$-1000-1000-$C$ on samples from the CIFAR10 [Krizhevsky et al., 2009] dataset. The model is trained to minimize the cross-entropy loss with full-batch gradient descent, once with two classes (250 samples per class) and once for the full 10 classes (50 samples per class). Both models train on the same amount of samples (500). The test set accuracy of the models is 77%/32% respectively, which is far from random (50%/10% resp.). See implementation details in Appendix B.

To quantify the quality of our reconstructed samples, for each sample in the original training set we search for its nearest neighbour in the reconstructed images and measure the similarity using SSIM [Wang et al., 2004] (higher SSIM means better reconstruction). In Fig. 2 we plot the quality of reconstruction (in terms of SSIM) against the distance of the sample from the decision boundary $\Phi_{y_i}(\mathbf{x}_i; \boldsymbol{\theta}) - \max_{j\neq y_i} \Phi_j(\mathbf{x}_i; \boldsymbol{\theta})$. As seen, a multi-class classifier yields much more samples that are vulnerable to being reconstructed.

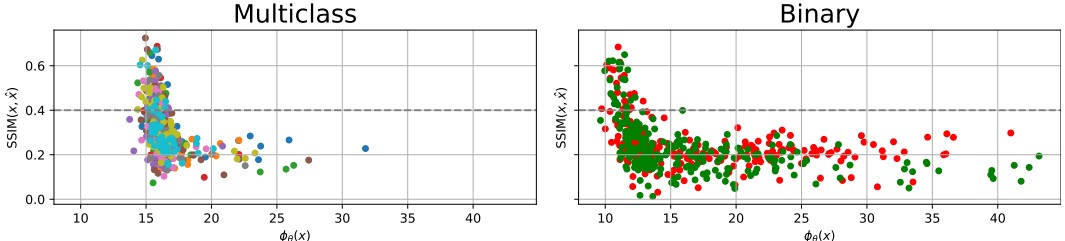

Figure 2: Multi-class classifiers are more vulnerable to training-set reconstruction. For a training set of size 500, a multi-class model (*left*) yields 101 reconstructed samples with good quality (SSIM>0.4), compared to 40 in a binary classification model (*right*).

We examine the relation between the ability to reconstruct from a model and the number of classes on which it was trained. Comparing between two models trained on different number of classes is not immediately clear, since we want to isolate the effect of the number of classes from the size of the dataset (it was observed by Haim et al. [2022] that the number of reconstructed samples decreases as the total size of the training set increases). We therefore train models on training sets with varying number of classes ($C \in \{2, 3, 4, 5, 10\}$) and varying number of samples per class $(1, 5, 10, 50)$. The results are visualized in Fig. 3a. As seen, for models with same number of samples per class, the ability to reconstruct *increases* with the number of classes, even though the total size of the training set is larger. This further validates our hypothesis that the more classes, the more samples are vulnerable to reconstruction (also see Appendix C).

Another way to validate this hypothesis is by showing the dependency between the number of classes and the number of "good" reconstructions (SSIM>0.4) – shown in Fig. 3b. As can be seen, training on multiple classes yields more samples that are vulnerable to reconstruction. An intuitive explanation, is that multi-class classifiers have more "margin-samples". Since margin-samples are more vulnerable to reconstruction, this results in more samples being reconstructed from the model.

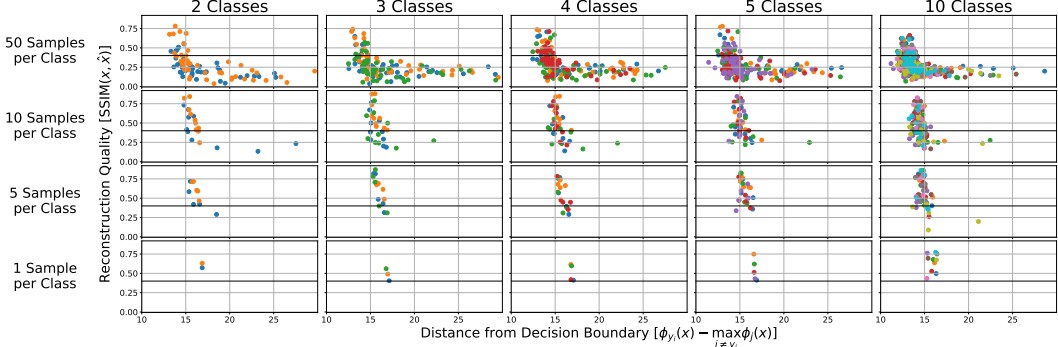

(a) Analysing the relation between number of classes and number of samples per class in reconstruction from multiclass Classifiers.

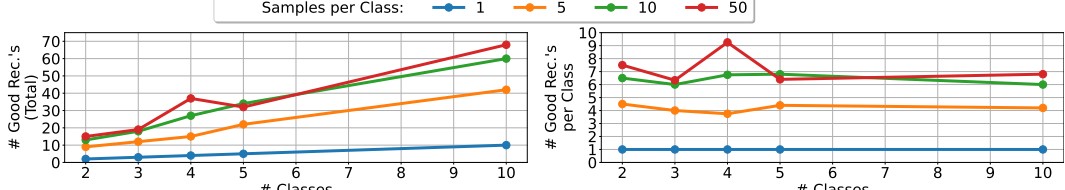

(b) Number of "good" reconstructions increases with number of classes and the samples per class

Figure 3: Evaluating the effect of multiple classes on the ability to reconstruct. We show reconstructions from models trained with different numbers of classes and different numbers of samples per class. As seen, multiple classes result in more reconstructed samples.

## 5 Data Reconstruction with General Loss Functions

We demonstrate that data reconstruction can be generalized to a larger family of loss functions. Haim et al. [2022] and Section 4 only considered a reconstruction scheme based on the implicit bias of gradient methods trained with cross-entropy loss. For other loss functions, such as the square loss, a precise characterization of the implicit bias in nonlinear networks does not exist [Vardi and Shamir, 2021]. Hence, we establish a reconstruction scheme for networks trained with explicit regularization, i.e., with weight decay. We show that as long as the training involves a weight-decay term, we can derive a reconstruction objective that is very similar to the previous objectives in Eqs. (6) and (12).

### 5.1 Theory

Let $\ell(\Phi(\mathbf{x}_i; \boldsymbol{\theta}), y_i)$ be a loss function that gets as input the predicted output of the model $\Phi$ (parametrized by $\boldsymbol{\theta}$) on an input sample $\mathbf{x}_i$, and its corresponding label $y_i$. The total regularized loss:

$$\mathcal{L} = \sum_{i=1}^{n} \ell(\Phi(\mathbf{x}_i; \boldsymbol{\theta}), y_i) + \lambda_{\text{WD}} \frac{1}{2} \|\boldsymbol{\theta}\|^2 \quad . \tag{13}$$

Assuming convergence ($\nabla_{\boldsymbol{\theta}} \mathcal{L} = 0$), the parameters should satisfy the following :

$$\boldsymbol{\theta} = \sum_{i=1}^{n} \ell_i' \, \nabla_{\boldsymbol{\theta}} \Phi(\mathbf{x}_i; \boldsymbol{\theta}) \tag{14}$$

where $\ell_i' = -\frac{1}{\lambda_{WD}} \frac{\partial \ell(\Phi(\mathbf{x}_i; \boldsymbol{\theta}), y_i)}{\partial \Phi(\mathbf{x}_i; \boldsymbol{\theta})}$. This form (which is similar to the condition in Eq. (2)), allows us to define a generalized reconstruction loss for models trained with a weight-decay term:

$$L_{rec}(\mathbf{x}_1, ..., \mathbf{x}_m, \lambda_1, ..., \lambda_m) = \|\boldsymbol{\theta} - \sum_{i=1}^{n} \lambda_i \nabla_{\boldsymbol{\theta}} \Phi(\mathbf{x}_i; \boldsymbol{\theta})\|_2^2 \tag{15}$$

As before, we also include the same $L_{\text{prior}}$ as in Section 3. It is straightforward to see that $L_{rec}$ is a generalization of the reconstruction loss in Eq. (6) ($y_i$ could be incorporated into the $\lambda_i$ term).

## 5.2   Results and Analysis

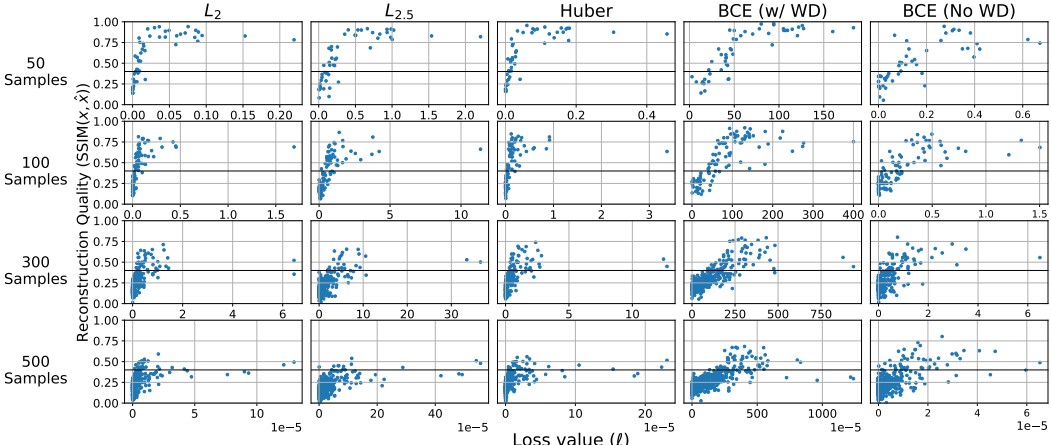

Figure 4: **Reconstruction from general losses** (column) for various training set sizes (row), using Eq. (15). "Harder" samples (with higher loss) are easier to reconstruct.

We validate the above theoretical analysis by demonstrating reconstruction from models trained on other losses than the ones shown in Section 4 and Haim et al. [2022]. We use the same dataset as in the classification tasks – images from CIFAR10 dataset with binary labels of $\{-1, 1\}$. The only difference is replacing the classification binary cross-entropy loss with regression losses (e.g., MSE).

In classification tasks, we analyzed the results by plotting the reconstruction quality (SSIM) against the sample's distance from the decision boundary (see Section 3). This showed that reconstruction is only feasible for margin-samples. However, in regression tasks, margin and decision boundary lack specific meaning. We propose an alternative analysis approach – note that smaller distance from the margin results in higher loss for binary cross-entropy. Intuitively, margin-samples are the most challenging to classify (as reflected by the loss function). Therefore, for regression tasks, we analyze the results by plotting the reconstruction quality against the loss (per training sample).

In Fig. 4 we show results for reconstructions from models trained with MSE, $L_{2.5}$ loss ($\ell = |\Phi(\mathbf{x}; \boldsymbol{\theta}) - y|^p$ for p=2,2.5 respectively) and Huber loss [Huber, 1992]. The reconstruction scheme in Eq. (15) is the same for all cases, and is invariant to the loss used during training. Fig. 4 highlights two important observations: first, the reconstruction scheme in Eq. (15) succeeds in reconstructing large portions of the training set from models trained with regression losses, as noted from the high

quality (SSIM) of the samples. Second, by plotting quality against the loss, one sees that "challenging" samples (with high loss) are easier to reconstruct. Also note that the analysis works for classification losses, namely BCE with or without weight-decay in Fig. 4). For more results see Appendix D.

# 6 On the Different Factors that Affect Reconstructability

Our goal is to gain a deeper understanding of the factors behind models' vulnerability to reconstruction schemes. In this section, we present several analyses that shed light on several important factors.

## 6.1 The Role of Weight Decay in Data Reconstruction

Haim et al. [2022] assumed MLP models whose first fully-connected layer was initialized with small (non-standard) weights. Models with standard initialization (e.g., He et al. [2015], Glorot and Bengio [2010]) did not yield reconstructed samples. In contrast, the MLPs reconstructed in Haim et al. [2022] were initialized with an extremely small variance in the first layer. Set to better understand this drawback, we observed that incorporating weight-decay during training, not only enabled samples reconstruction in models with standard initialization, but often increase the reconstructability of training samples.

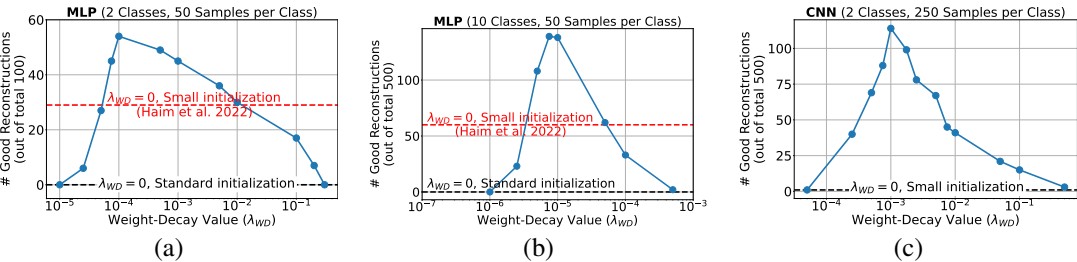

Figure 5: Using weight-decay during training increases vulnerability to sample reconstruction

In Fig. 5a-b we show the number of good reconstructions for different choices of the value of the weight decay ($\lambda_{\text{WD}}$), for MLP classifiers trained on $C$=2,10 classes and 50 samples per class (Fig. 5 a, b resp.). We add two baselines trained *without* weight-decay: model trained with standard initialization (black) and model with small-initialized first-layer (red). See how for some values of weight-decay, the reconstructability is *significantly higher* than what was observed for models with non-standard initialization. By examining the training samples' distance to the boundary, one observes that using weight-decay results in more margin-samples which are empirically more vulnerable to reconstruction (see full details in Appendix E).

We now give an intuitive theoretical explanation to the role of weight decay in data reconstruction. Theorem 3.1 is used to devise the reconstruction loss in Eq. (6), which is based on the directional convergence to a KKT point of the max-margin problem. However, this directional convergence occurs asymptotically as the time $t \to \infty$, and the rate of convergence in practice might be extremely slow. Hence, even when training for, e.g., $10^6$ iterations, gradient descent might reach a solution which is still too far from the KKT point, and therefore reconstruction might fail. Thus, even when training until the gradient of the empirical loss is extremely small, the direction of the network's parameters might be far from the direction of a KKT point. In Moroshko et al. [2020], the authors proved that in *diagonal linear networks* (i.e., a certain simplified architecture of deep linear networks) the initialization scale controls the rate of convergence to the KKT point, namely, when the initialization is small gradient flow converges much faster to a KKT point. A similar phenomenon seems to occur also in our empirical results: when training without weight decay, small initialization seems to be required to allow reconstruction. However, when training with weight decay, our theoretical analysis in Section 5.1 explains why small initialization is no longer required. Here, the reconstruction does not rely on converging to a KKT point of the max-magin problem, but relies on Eq. (14) which holds (approximately) whenever we reach a sufficiently small gradient of the training objective. Thus, when training with weight decay and reaching a small gradient Eq. (14) holds, which allows for reconstruction, contrary to training without weight decay where reaching a small gradient does not imply converging close to a KKT point.

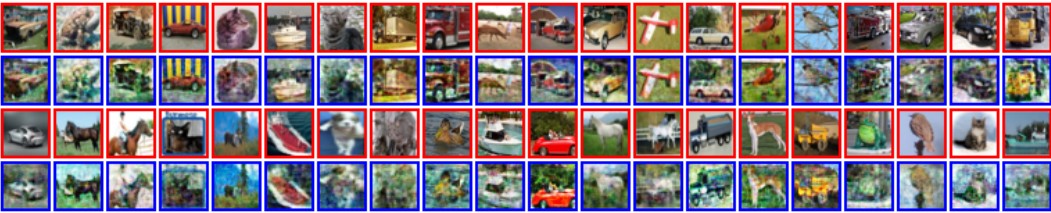

Figure 6: **Reconstruction from CNN.** Training samples (red) and their best reconstructions (blue)

**Reconstruction from Convolutional Neural Networks (CNNs).** CNNs adhere to the assumptions of Theorem 3.1, yet Haim et al. [2022] failed to apply their reconstruction scheme Eq. (6) to CNNs. We observe that incorporating weight-decay during training (using standard initialization) enables samples reconstruction. In Fig. 6 we show an example for reconstruction from a binary classifier whose first layer is a Conv-layer with kernel size 3 and 32 output channels, followed by two fully connected layers (CONV($k$=3,$C_{\text{out}}$=32)-1000-1). The weight-decay term is $\lambda_{\text{WD}}$=0.001 (the training setup is similar to that of MLP). In Fig. 5c we show the reconstructability for the same CNN model trained with other values of $\lambda_{\text{WD}}$. Note how the weight-decay term plays similar role in the CNN as in the MLP case. See full details in Appendix F.

## 6.2 The Effect of the Number of Parameters and Samples on Reconstructability

Haim et al. [2022] observed that models trained on fewer samples are more susceptible to reconstruction in terms of both quantity and quality. In this section, we delve deeper into this phenomenon, focusing on the intricate relationship between the number of parameters in the trained model and the number of training samples. We conduct the following experiment:

We train 3-layer MLPs with architecture $D$-$W$-$W$-1 on $N$ training samples from binary CIFAR10 (animals vs. vehicles), where $W \in \{5, 10, 50, 100, 500, 1000\}$ and $N \in \{10, 50, 100, 300, 500\}$. We conduct the experiment for both classification and regression losses, with BCE and MSE loss respectively. Generalization error is 23%-31% for BCE (classification) and 0.69-0.88 for MSE (regression), compared to 50%/0.97 for similar models with random weights.

We reconstruct each model using Eq. (15) and record the number of good reconstructions. The results are shown in Fig. 7. Note that as $W/N$ increases, our reconstruction scheme is capable of reconstructing more samples, and vice versa. For example, consider the case when $N$=10. To successfully reconstruct the entire training set, it is sufficient for $W$ to be greater than 50/10 (for MSE/BCE). However, when $N$=500, even larger models (with larger $W$) can only reconstruct up to 8% of the samples.

Lastly, we reconstruct from a model with $W$=10,000, trained on $N$=5,000 samples (5 times larger than any previous model). While there is some degradation in the quality of the reconstructions compared to models trained on fewer samples, it is evident that our scheme can still reconstruct some of the training samples. For full results see Appendix G.

## 7 Conclusions

We present improved reconstruction methods and conduct a comprehensive analysis of the reconstruction method proposed by Haim et al. [2022]. Particularly, we extend their reconstruction scheme to a multi-class setting and devise a novel reconstruction scheme for general loss functions, allowing reconstruction in a regression setting (e.g., MSE loss). We examine various factors influencing reconstructability. We shed light on the role of weight decay in samples memorization, allowing for sample reconstruction from convolutional neural networks. Lastly, we examine the intricate relationship between the number of parameters, the number of samples, and the vulnerability of the model to reconstruction schemes. We acknowledge that our reconstruction method raises concerns regarding privacy. We consider it crucial to present such methodologies as they encourage researchers to study the potential hazards associated with training neural networks. Additionally, it allows for the development of protective measures aimed at preventing the leakage of sensitive information.

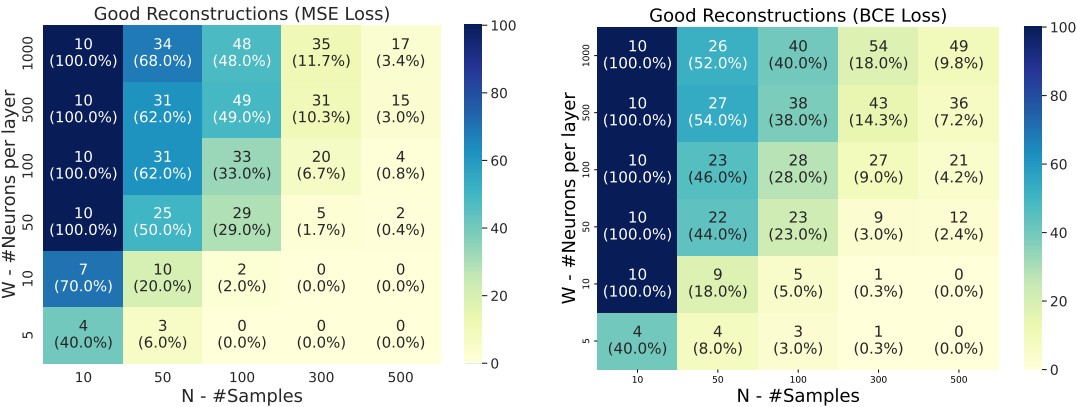

Figure 7: **Effect of the number of neurons and number of training samples on reconstructability.** We train 3-layer MLPs with architecture $D$-$W$-$W$-$1$ on $N$ training samples from binary CIFAR10 (animals vs. vehicles), using MSE (*left*) or BCE (*right*) loss. At each cell we report the number of good reconstructions (SSIM>0.4), in both absolute numbers and as a percentage relative to $N$.

**Limitations.**    While we have relaxed several of the previous assumptions presented in Haim et al. [2022], our method still exhibits certain limitations. First, we only consider relatively small-scale models: up to several layers, without residual connections, that have trained for many iterations on a relatively small dataset without augmentations. Second, determining the optimal hyperparameters for the reconstruction scheme poses a challenge as it requires exploring many configurations for even a single model.

**Future work.**    All of the above extend our knowledge and understanding of how memorization works in certain neural networks. This opens up several possibilities for future research including extending our reconstruction scheme to practical models (e.g., ResNets), exploring reconstruction from models trained on larger datasets or different data types (e.g., text, time-series, tabular data), analyzing the impact of optimization methods and architectural choices on reconstructability, and developing privacy schemes to protect vulnerable samples from reconstruction attacks.

**Acknowledgements.**    This project received funding from the European Research Council (ERC) under the European Union's Horizon 2020 research and innovation programme (grant agreement No 788535), and ERC grant 754705, and from the D. Dan and Betty Kahn Foundation. GV acknowledges the support of the NSF and the Simons Foundation for the Collaboration on the Theoretical Foundations of Deep Learning.

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

# A   Analyzing and Visualizing the Results of the Reconstruction Optimization

The analysis of the results of the various reconstruction losses Eqs. (6), (12) and (15), involve verifying and checking which of the training samples were reconstructed. In this section we provide further details on our method for analyzing the reconstruction results, and how we measure the quality of our reconstructions.

## A.1   Analyzing the Results of the Reconstruction Optimization

In order to match between samples from the training set and the outputs of the reconstruction algorithm (the so-called "candidates") we follow the same protocol of Haim et al. [2022]. Note that before training our models, we subtract the mean image from the given training set. Therefore the training samples are $d$-dimensional objects where each entry is in $[-1, 1]$.

First, for each training sample we compute the distance to all the candidates using a normalized $L_2$ score:

$$d(\mathbf{x}, \mathbf{y}) = \left\| \frac{\mathbf{x} - \mu_\mathbf{x}}{\sigma_\mathbf{x}} - \frac{\mathbf{y} - \mu_\mathbf{y}}{\sigma_\mathbf{y}} \right\|_2^2 \tag{16}$$

Where $\mathbf{x}, \mathbf{y} \in \mathbb{R}^d$ are a training sample or an output candidate from the reconstruction algorithm, $\mu_\mathbf{x} = \frac{1}{d} \sum_{i=1}^d \mathbf{x}(i)$ is the mean of $\mathbf{x}$ and $\sigma_\mathbf{x} = \sqrt{\frac{1}{d-1} \sum_{i=1}^d (\mathbf{x}(i) - \mu_\mathbf{x})^2}$ is the standard deviation of $\mathbf{x}$ (and the same goes for $\mathbf{y}, \mu_\mathbf{y}, \sigma_\mathbf{y}$).

Second, for each training sample, we take $C$ candidates with the smallest distance according to Eq. (16). $C$ is determined by finding the first candidate whose distance is larger than $B$ times the distance to the closest nearest neighbour (where $B$ is a hyperparameter). Namely, for a training sample $\mathbf{x}$, the nearest neighbour is $\mathbf{y}_1$ with a distance $d(\mathbf{x}, \mathbf{y}_1)$, then $C$ is determined by finding a candidate $\mathbf{y}_{C+1}$ whose distance is $d(\mathbf{x}, \mathbf{y}_{C+1}) > B \cdot d(\mathbf{x}, \mathbf{y}_1)$, and for all $j \leq C, d(\mathbf{x}, \mathbf{y}_j) \leq B \cdot d(\mathbf{x}, \mathbf{y}_1)$. $B$ was chosen heuristically to be $B = 1.1$ for MLPs, and $B = 1.5$ for convolutional models. The $C$ candidates are then summed to create the reconstructed sample $\hat{\mathbf{x}} = \frac{1}{C} \sum_{j=1}^C \mathbf{y}_j$. In general, we can also take only $C = 1$ candidate, namely just one nearest neighbour per training sample, but choosing more candidates improve the visual quality of the reconstructed samples.

Third, the reconstructed sample $\hat{\mathbf{x}}$ is scaled to an image in $[0, 1]$ by adding the training set mean and linearly "stretching" the minimal and maximal values of the result to $[0, 1]$. Finally, we compute the SSIM between the training sample $\mathbf{x}$ and the reconstructed sample $\hat{\mathbf{x}}$ to measure the quality of reconstruction.

## A.2   Deciding whether a Reconstruction is "Good"

Here we justify our selection for SSIM=0.4 as the threshold for what we consider as a "good" reconstruction. In general, the problem of deciding whether a reconstruction is the correct match to a given sample, or whether a reconstruction is a "good" reconstruction is equivalent to the problem of comparing between images. No "synthetic" metric (like SSIM, $l2$ etc.) will be aligned with human perception. A common metric for this purpose is LPIPS [Zhang et al., 2018] that uses a classifier trained on Imagenet [Deng et al., 2009], but since CIFAR images are much smaller than Imagenet images ($32 \times 32$ vs. $224 \times 224$) it is not clear that this metric will be better than SSIM.

As a simple rule of thumb, we use SSIM>0.4 for deciding that a given reconstruction is "good". To justify, we plot the best reconstructions (in terms of SSIM) in Fig. 8. Note that almost all samples with SSIM>0.4 are also visually similar (for a human). Also note that some of the samples with SSIM<0.4 are visually similar, so in this sense we are "missing" some good reconstructions. In general, determining whether a candidate output of a reconstruction algorithm is a match to a training sample is an open question and a problem in all other works for data reconstruction, see for example Carlini et al. [2023] that derived a heuristic for reconstructed samples from a generative model. This cannot be dealt in the scope of this paper, and is an interesting future direction for our work.

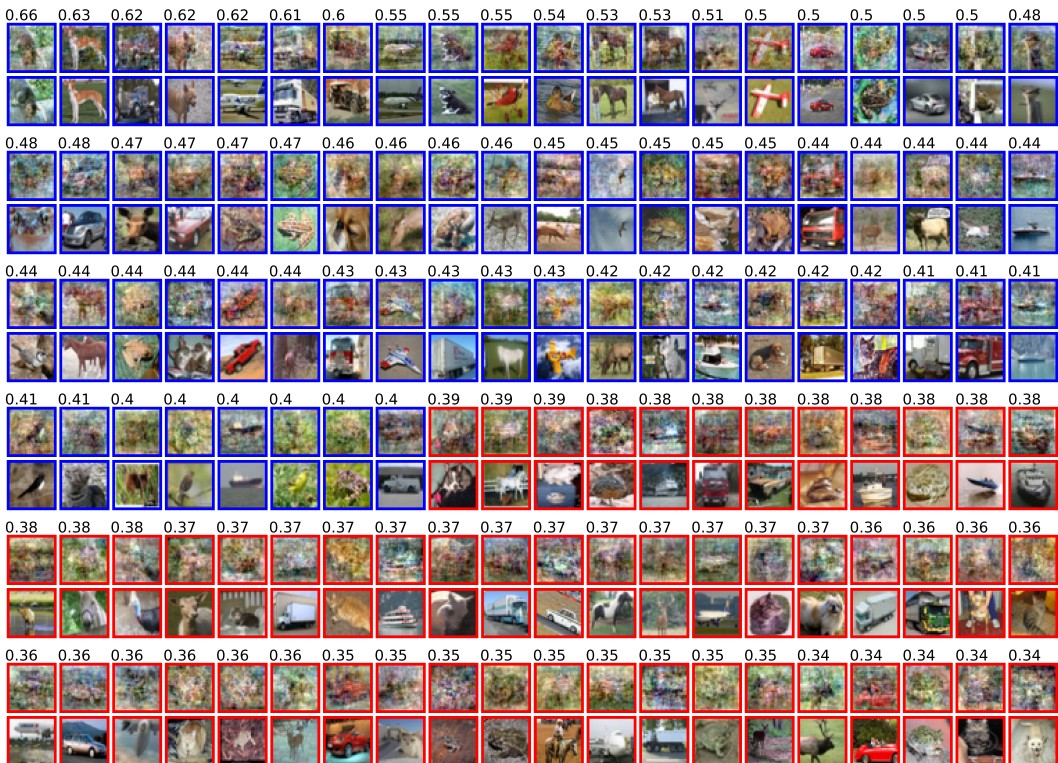

Figure 8: Justifying the threshold of SSIM= $0.4$ as good rule-of-thumb for a threshold for a "good" reconstruction. The SSIM values are shown above each train-reconstruction pair. Note that samples with SSIM> $0.4$ (blue) are visually similar. Also some of the samples with SSIM< $0.4$ (red) are similar. In general deciding whether a reconstruction is "good" is an open question beyond the scope of this paper.

## B    Implementation Details

**Further Training Details.**    The models that were reconstructed in the main part of the paper were trained with learning rates of $0.01$ for binary classifiers (both MLP and convolutional), and $0.5$ in the case of multi-class classifier (Section 4). The models were trained with full batch gradient descent for $10^6$ epochs, to guarantee convergence to a KKT point of Eq. (1) or a local minima of Eq. (13). When small initialization of the first layer is used (e.g., in Figs. 2 and 3), the weights are initialized with a scale of $10^{-4}$. We note that Haim et al. [2022] observed that models trained with SGD can also be reconstructed. The experiment in Appendix G (large models with many samples) also uses SGD and results with similar conclusion, that some models trained with SGD can be reconstructed. In general, exploring reconstruction from models trained with SGD is an interesting direction for future works.

**Runtime and Hardware.**    Runtime of a single reconstruction run (specific choice of hyperparameters) from a model $D$-1000-1000-1 takes about 20 minutes on a GPU Tesla V-100 32GB or NVIDIA Ampere Tesla A40 48GB.

**Hyperparameters of the Reconstruction Algorithm.**    Note that the reconstruction loss contains the derivative of a model with ReLU layers, which is flat and not-continuous. Thus, taking the derivative of the reconstruction loss results in a zero function. To address this issue we follow a solution presented in Haim et al. [2022]. Namely, given a trained model, we replace in the backward phase of backpropogation the ReLU function with the derivative of a softplus function (or SmoothReLU) $f(x) = \alpha \log(1 + e^{-x})$, where $\alpha$ is a hyperparameter of the reconstruction scheme. The functionality of the model itself does not change, as in the forward phase the function remains a ReLU. Only the backward function is replaced with a smoother version of the derivative of ReLU which is $f'(x) = \alpha\sigma(x) = \frac{\alpha}{1+e^{-x}}$ (here $\sigma$ is the Sigmoid function). To find good reconstructions we run the

algorithm multiple times (typically 100 times) with random search over the hyperparameters (using the Weights & Biases framework [Biewald, 2020]). The exact parameters for the hyperparameters search are:

- Learning rate: log-uniform in $[10^{-5}, 1]$
- $\sigma_x$: log-uniform in $[10^{-6}, 0.1]$
- $\lambda_{\min}$: uniform in $[0.01, 0.5]$
- $\alpha$: uniform in $[10, 500]$

## C  Multiclass Reconstruction - More Results

### C.1  Experiments with Different Number of Classes and Fixed Training Set Size

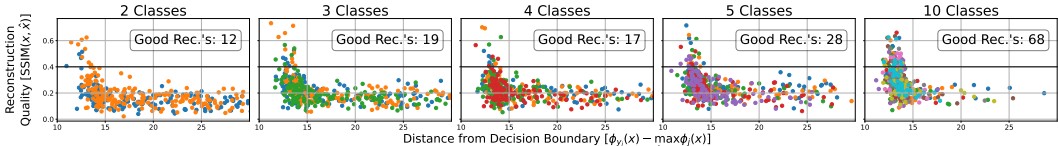

Figure 9: Experiments of reconstruction from models trained on a a fixed training set size (500 samples) for different number of classes. Number of "good" reconstruction is shown for each model.

To complete the experiment shown in Fig. 3, we also perform experiments on models trained on various number of classes ($C \in \{2, 3, 4, 5, 10\}$) and with a fixed training set size of 500 samples (distributed equally between classes), see Fig. 9. It can be seen that as the number of classes increases, also does the number of good reconstructions, where for 10 classes there are more than 6 times good reconstructions than for 2 classes. Also, the quality of the reconstructions improves as the number of classes increase, which is depicted by an overall higher SSIM score. We also note, that the number of good reconstructions in Fig. 9 is very similar to the number of good reconstructions from Fig. 3 for 50 samples per class. We hypothesize that although the number of training samples increases, the number of "support vectors" (i.e samples on the margin which can be reconstructed) that are required for successfully interpolating the entire dataset does not change by much.

### C.2  Results on SVHN Dataset

As shown in Fig. 10, our multiclass reconstruction scheme is not limited to CIFAR10 dataset, but can be extended to other datasets as well, specifically SVHN dataset [Netzer et al., 2011]. The model whose reconstructions are shown in Fig. 10 was trained on 50 samples per class (total of 10 classes) and the rest of the training hyperparameters are the same as that of its CIFAR10 equivalent (of 50 sample per class with 10 classes).

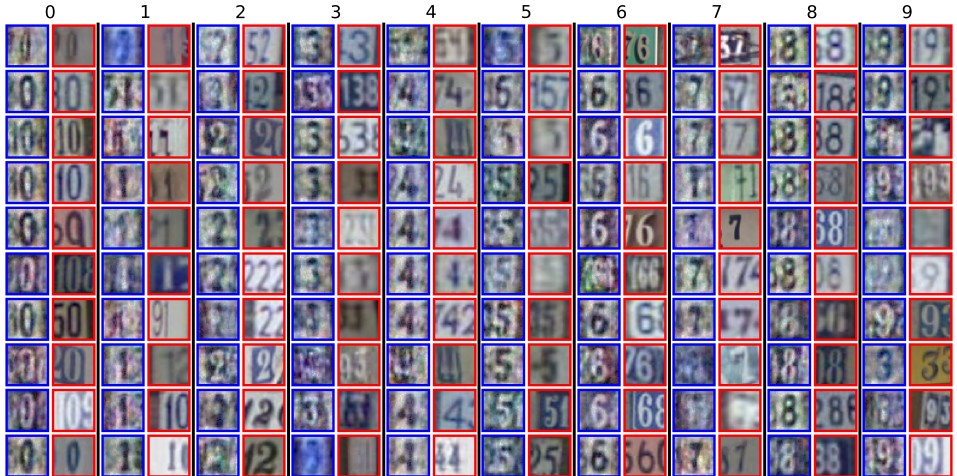

Figure 10: Reconstruction form model trained on 50 samples per class from the SVHN dataset.

## D    General Losses - More Results

Following the discussion in Section 5 and Fig. 4, Figures 11, 12, 13 present visualizations of training samples and their reconstructions from models trained with $L_2$, $L_{2.5}$ and Huber loss, respectively.

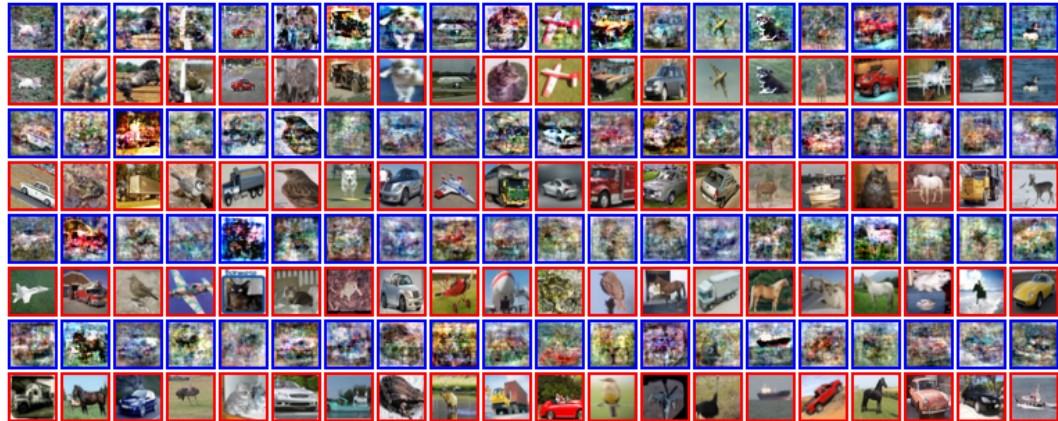

Figure 11: **Reconstruction using $L_2$ loss.** Training samples (red) and their best reconstructions (blue) using an MLP classifier that was trained on 300 CIFAR10 images using an $L_2$ regression loss, as described in Section 5 and Fig. 4.

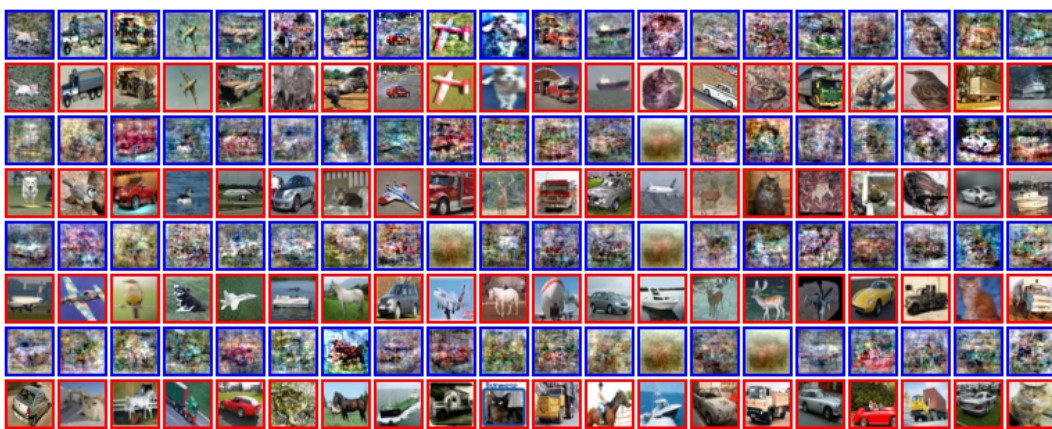

Figure 12: **Reconstruction using $L_{2.5}$ loss.** Training samples (red) and their best reconstructions (blue) using an MLP classifier that was trained on 300 CIFAR10 images using an $L_{2.5}$ regression loss, as described in Section 5 and Fig. 4.

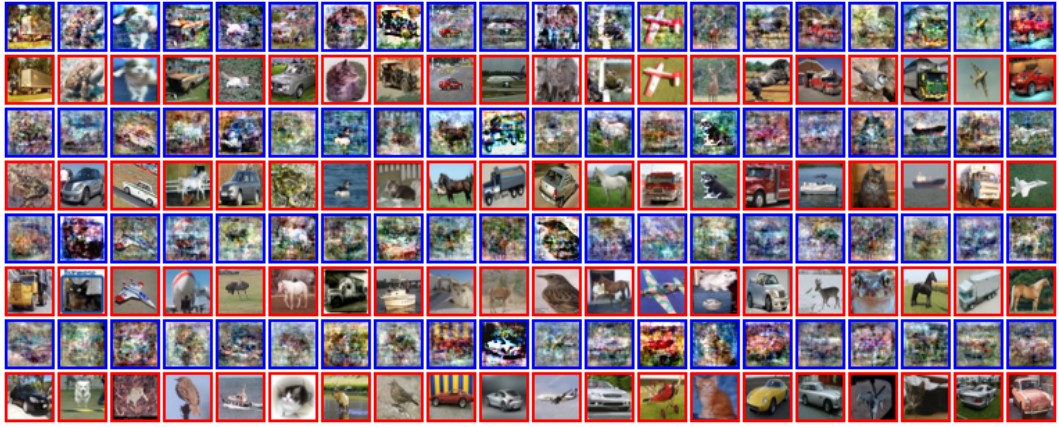

Figure 13: **Reconstruction using Huber loss.** Training samples (red) and their best reconstructions (blue) using an MLP classifier that was trained on 300 CIFAR10 images using Huber loss, as described in Section 5 and Fig. 4.

## E   Further Analysis of Weight Decay

By looking at the exact distribution of reconstruction quality to the distance from the margin, we observe that weight-decay (for some values) results in more training samples being on the margin of the trained classifier, thus being more vulnerable to our reconstruction scheme.

This observation is shown in Fig. 14 where we show the scatter plots for all the experiments from Fig. 5 (a). We also provide the train and test errors for each model. It seems that the test error does not change significantly. However, an interesting observation is that reconstruction is possible even for models with non-zero training errors, i.e. models that do not interpolate the data, for which the assumptions of Lyu and Li [2019] do not hold.

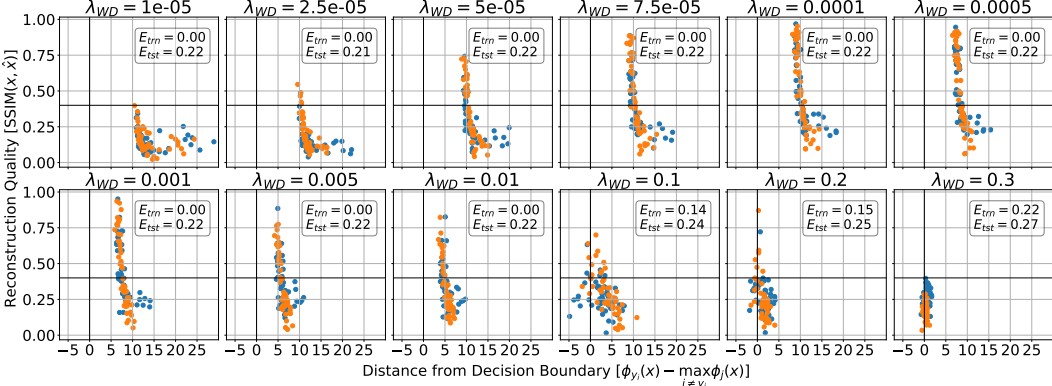

Figure 14: Scatter plots of the 12 experiments from Fig. 5 (a). Each plot is model trained with a different value of weight decay on 2 classes with 50 samples in each class. Certain values of weight decay make the model more susceptible to our reconstruction scheme.

## F   Convolutional Neural Networks - Ablations and Observations

In this section we provide more results and visualizations to the experiments on convolutional neural network in Section 6.1.

In Fig. 15 we show ablations for the choice of the kernel-size ($k$) and number of output channels ($C_{\text{out}}$) for models with architecture CONV(kernel-size=$k$,output-channels=$C_{\text{out}}$)-1000-1. All models were trained on 500 images (250 images per class) from the CIFAR10 dataset, with weight-decay

term $\lambda_{\mathrm{WD}}$=0.001. As can be seen, for such convolutional models we are able to reconstruct samples for a wide range of choices.

Note that the full summary of reconstruction quality versus the distance from the decision boundary for the model whose reconstucted samples are shown in Fig. 6, is shown in Fig. 15 for kernel-size 3 (first row) and number of output channels 32 (third column).

**Further analysis of Fig. 15.** As expected for models with less parameters, the reconstructability decreases as the number of output channels decrease. An interesting phenomenon is observed for varying the kernel size: for a fixed number of output channel, as the kernel size increases, the susceptibility of the model to our reconstruction scheme decreases. However, as the kernel size approaches 32 (the full resolution of the input image), the reconstructability increases once again. On the one hand it is expected, since for kernel-size=32 the model is essentially an MLP, albeit with smaller hidden dimension than usual (at most 64 here, whereas the typical model used in the paper had 1000). On the other hand, it is not clear why for some intermediate values of kernel size (in between 3 and 32) the reconstructability decreases dramatically (for many models there are no reconstructed samples at all). This observation is an interesting research direction for future works.

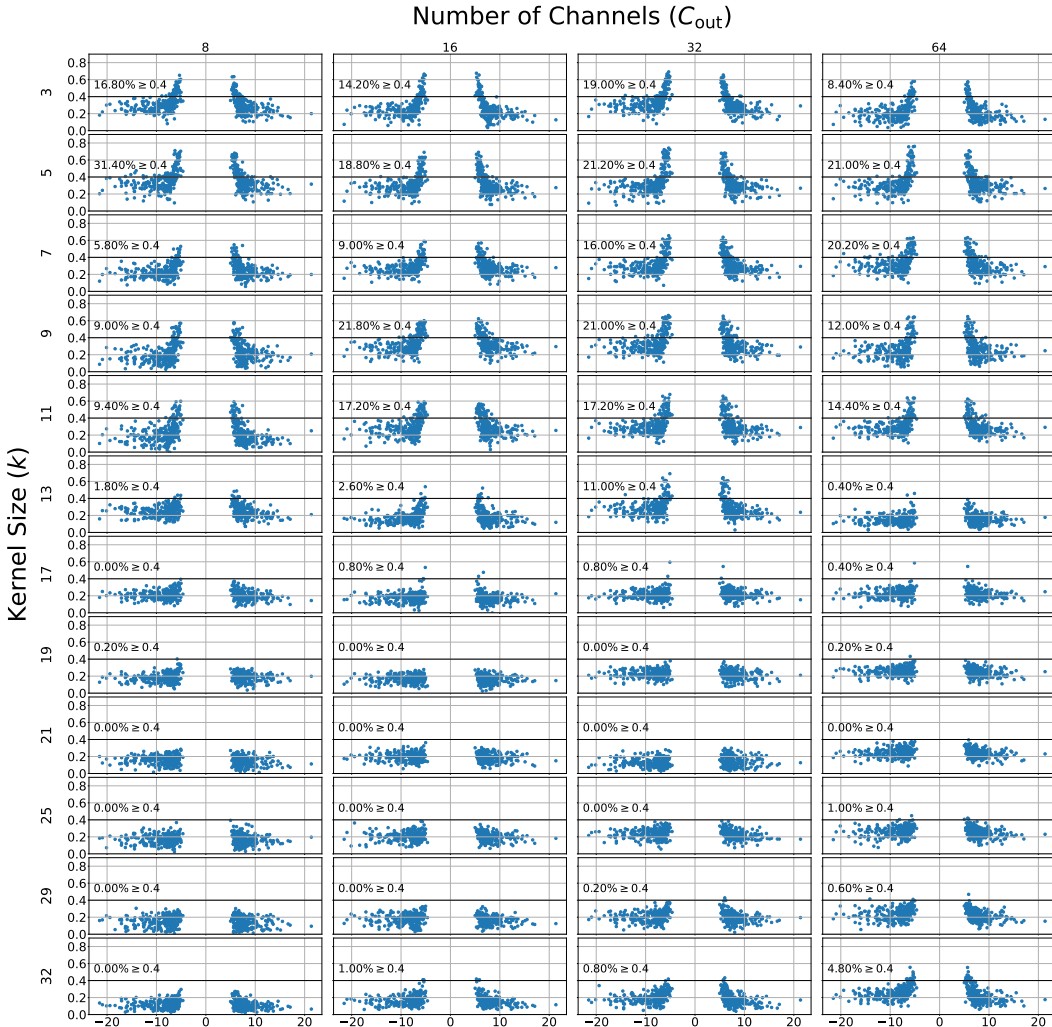

Figure 15: **Ablating the choice of the kernel size and output-channels** for reconstruction from neural binary classifiers with architecture CONV(kernel-size=$k$,output-channels=$C_{\mathrm{out}}$)-1000-1. (Please note that this figure might not be displayed well on Google Chrome. Please open in Acrobat reader.)

**Visualizing Kernels.**  In Haim et al. [2022], it was shown that some of the training samples can be found in the first layer of the trained MLPs, by reshaping and visualizing the weights of the first fully-connected layer. As opposed to MLPs, in the case of a model whose first layer is a convolution layer, this is not possible. For completeness, in Fig. 16 we visualize all 32 kernels of the Conv layer. Obviously, full images of shape 3x32x32 cannot be found in kernels of shape 3x3x3, which makes reconstruction from such models (with convolution first layer) even more interesting.

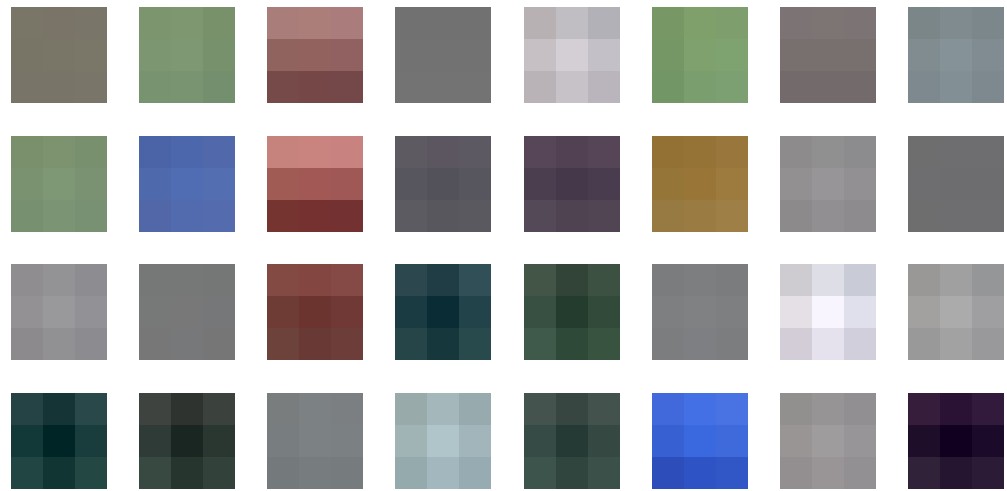

Figure 16: The kernels of the model whose reconstructions are shown in Fig. 6, displayed as RGB images.

## G    Reconstruction From a Larger Number of Samples

One of the major limitations of Haim et al. [2022] is that they reconstruct from models that trained on a relatively small number of samples. Specifically, in their largest experiment, a model is trained with only 1,000 samples. Here we take a step further, and apply our reconstruction scheme for a model trained on 5,000 data samples.

To this end, we trained a 3-layer MLP, where the number of neurons in each hidden layer is 10,000. Note that the size of the hidden layer is 10 times larger than in any other model we used. Increasing the number of neurons seems to be one of the major reasons for which we are able to reconstruct from such large datasets, although we believe it could be done with smaller models, which we leave for future research. We used the CIFAR100 dataset, with 50 samples in each class, for a total of 5000 samples.

In Fig. 17a we give the best reconstructions of the model. Note that although there is a degradation in the quality of the reconstruction w.r.t a model trained on less samples, it is still clear that our scheme can reconstruct some of the training samples to some extent. In Fig. 17b we show a scatter plot of the SSIM score w.r.t the distance from the boundary, similar to Fig. 3a. Although most of the samples are on or close to the margin, only a few dozens achieve an SSIM$> 0.4$. This may indicate that there is a potential for much more images to reconstruct, and possibly with better quality.

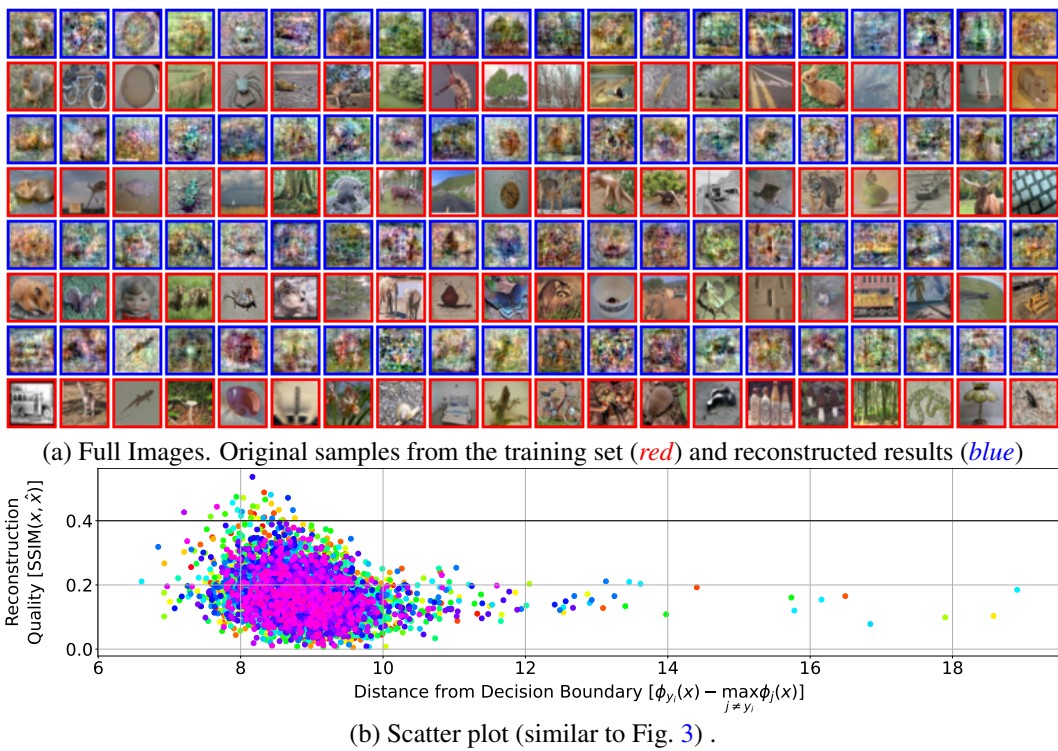

(a) Full Images. Original samples from the training set (*red*) and reconstructed results (*blue*)

(b) Scatter plot (similar to Fig. 3) .

Figure 17: Reconstruction from a model trained on 50 images per class from the CIFAR100 dataset (100 classes, total of 5000 datapoints). The model is a 3-layer MLP with 10000 neurons in each layer. (Please note that this figure might not be displayed well on Google Chrome. Please open in Acrobat reader.)

