# OpenReview forum: "Deconstructing Data Reconstruction: Multiclass, Weight Decay and General Losses"
_NeurIPS.cc/2023/Conference — NeurIPS 2023 poster_

### Official Review · Reviewer_nrvg · 2023-06-23

**Soundness:** 3 good
**Presentation:** 4 excellent
**Contribution:** 3 good
**Rating:** 7
**Confidence:** 3

**Summary:**

The paper presents multiple extensions of Haim et al. [2022] in relation to reconstruction of training data in neural networks. In short the extensions are: extension to multiclass from binary, extension to regression losses, investigate of the effect of weight decay and analysis of relationship between number of samples and number of parameters. All extensions are supported by experiments and for the first two extensions the authors provide theoretical foundation.

**Strengths:**

Originality:
The work seems original and novel (even if it is only incremental)

Quality:
The paper is of high quality and provide theoretical foundation where it is needed and empirical evidence for everything else.

Clarity:
The paper is clearly written and easy to follow.

Significance:
It is not unreasonable to say that the end goal for this line of work is to be able to study neural networks that are used in more real life scenarios today. In that regard this works provides an important stepping stone in that direction, especially with its extension to multiclass.

**Weaknesses:**

Section 5 on general loss functions seems to be the weaker contribution primarily due to two reasons:
1. The authors are essentially trying out multiple regression losses, which is good, but selling this as "general loss functions" seems to be a stretch. The authors could be more clear in their contributions by simply calling "general regressions losses".
2. The experiment is created by artificially taking a task that is binary in nature (labels {-1,1}) and trying to model them using a regression loss. It would be much more relevant if the authors had considered another dataset and/or task where the target is continues.

In general I find that the paper has a lot of experiments to show and due to the limited space this does not give room for an actual discussion of their findings. I find this a wrong prioritization, and that the authors should discuss and give some kind of reasoning for just some of their findings (see questions).

**Questions:**

The empirical results are impressive and speaks for themself, but I am really missing a bit of discussion of the results:
* Any reasoning why the straightforward extension failed to reconstruct samples (L139-141)?
* The reason why CNNs are harder to reconstruct? Essentially they are just sparse linear layers (compared to fully connected)
* Any reasoning why weight-decay can have such an influence on the number of reconstructed examples.

**Limitations:**

It is clear from the paper that work on reconstruction of training data from memorization in the neural networks are still in its early stages. That said it would be great if the authors could be more explicit in the paper about the obvious limitations of the method:
* Only possible for shallow networks
* Only possible for specific networks (only ReLU it seems)
* Only possible for networks trained on few samples (the extension to 5000 samples is nice, but still nowhere near practical settings)

It is mentioned in future work, but giving a bit of reasoning what challenges lies in scaling the methods seems to be in order.

---

> ### Author Rebuttal · Authors · 2023-08-09
>
> We thank the reviewer for the thorough review and constructive and positive feedback.
>
> Regrading general vs. regression loss: we agree with the reviewer on the comment. Although our method described in Section 5 allows us to reconstruct data which is trained using any loss function, we tested it only on several regression losses. We will change the phrasing throughout the paper to "general regression losses" instead of "general loss functions".
> Regarding datasets with continuous labels, we agree that these are interesting cases to study, however it is beyond the scope of this paper and we leave it to future research.
>
> Regarding the prioritization, we agree it would be helpful to provide more discussion on the findings and theoretical insights (e.g. such as in sections 4.1 and 5.1), however we had to trade-off in terms of space limitations to also present the results and figures in the main part of the paper. In the camera-ready version we will add more discussion on the findings themselves, and provide more details on the theoretical reasoning behind our results. Please also see the response to the last question below for further justification for weight decay.
>
>
> Regarding the questions:
>
> 1) This is a good point and we will further elaborate on it in the camera-ready version. The straightforward extension has $C \cdot n$ constraints whereas our proposed equivalent formulation has only $n$. For reconstruction, each such constraint requires its own optimization variable $\lambda$.
> So the latter formulation has much less optimization variables ($n$ < $C\cdot n$) which is very significant for large number of classes $C$. We believe that reducing the number of constraints is the main reason for the successful reconstruction in the multi-class setting.
>
> 2) Although CNNs can be written as a sparse linear layer, they are inherently different. First, there is weight sharing, and second, the number of parameters is significantly smaller than linear layers. For these reasons, the previous approach from [Haim et al. 2022] did not work as is. But we could reconstruct from CNNs using weight decay and large enough layers.
>
> 3) This is a very good question. We do have a theoretical justification for why weight decay helps reconstructability, at least for simplified networks. We will add this explanation in the camera-ready version. In a nutshell,
> Theorem 3.1 from our paper is used to devise a reconstruction loss which is based on that networks converge in direction to a  KKT point of the max-margin problem. However, this directional convergence occurs asymptotically as the time $t \to \infty$, and the rate of convergence in practice might be extremely slow, logarithmic in $t$. Hence, even when training for, e.g., $10^6$ iterations, gradient descent might reach a solution which is still too far from the KKT point, and therefore reconstruction fails. In other words, even when training until the gradient of the empirical loss is extremely small, the direction of the network’s parameters might be far from the direction of a KKT point. In [1], the authors proved that in diagonal linear networks (i.e., a certain simplified architecture of deep linear networks) the initialization scale indeed controls the rate of convergence to the KKT point, namely, when the initialization is small gradient flow converges much faster to a KKT point.
>
>     This theoretical result explains the behavior that we observe in the experiments, for a simple setting. For this reason, when training without weight decay, small initialization seems to be required to allow reconstruction. However, when training with weight decay, our theoretical analysis in Section 5.1 explains why small initialization is no longer required. Here, the reconstruction does not rely on converging to a KKT point of the max-magin problem, but relies on Eq. (14) which holds (approximately) whenever we reach a sufficiently small gradient of the training objective. 	Thus, when training with weight decay and reaching a small gradient Eq. (14) holds, which allows for reconstruction, contrary to training without weight decay where reaching a small gradient doesn't imply converging close to a KKT point.
>
> Regarding limitations, we will add a dedicated limitations section in the camera-ready version with elaborated discussion on the points highlighted by the reviewer.

---

> > ### Comment · Reviewer_nrvg · 2023-08-15
> > **Response to authors**
> >
> > I thank the authors for their long and elaborated answer. In my initial review I was mostly unhappy with the missing discussion of results, but based on the authors feedback it is clear that they have indeed thought about their method. I am hoping most of this discussion will go into the camera-ready version of the paper, even if some of the results should be moved to appendix.
> >
> > I am still positive regarding the paper, and even accounting for some of the other reviews, I will be keeping my score.

---

### Official Review · Reviewer_FQAU · 2023-07-03

**Soundness:** 3 good
**Presentation:** 3 good
**Contribution:** 3 good
**Rating:** 5
**Confidence:** 3

**Summary:**

Reconstructing training samples from the trained model may cause privacy issues. This paper demonstrates the training samples reconstruction for multi-classes and also for both fully-connected neural networks and convolutional neural networks. In addition, the authors investigate different factors that contribute the reconstruction.

**Strengths:**

1) Extended the previous training-sample reconstruction scheme to multi-class settings and also to both fully-connected NN and CNN, making it more practical.
2) It includes both theoretical analysis and empirical results to verify its effectiveness.

**Weaknesses:**

1) When the authors reconstruct the data, the generalization of the models is very poor. However, in practice, the trained model may also do well on unseen test data. So, the question is, can we still reconstruct the data successfully even when the trained model performs well on unseen test data?

2) What if a model is trained without weight decay or a model trained with other regularizers? Will this finding still be held?

3) The experimental datasets are relatively simple and on a small scale. Will it be possible to experiment on a more large-scale and complex dataset such as ImageNet?



#---------------------------------------------------------------------------------------------------------------#

#---------------------------------------------------------------------------------------------------------------#


 After reading the rebuttals and the reviews from other reviewers and the discussions, some of my previous concerns of this paper remain unaddressed, therefore, I would like to stick to my original evaluation.

**Questions:**

See weaknesses.

**Limitations:**

Did not discuss the limitation.

---

> ### Author Rebuttal · Authors · 2023-08-08
>
> We thank the reviewer for the comments and feedback. Regarding the questions raised by the reviewer:
>
> 1) Our work considers relatively small-scale networks, trained on  small datasets (up to $5$,$000$ samples). Although the generalization of our models is not comparable to the state of art, it is significantly non-trivial (much larger than random). We also ran an extensive hyper-parameter sweep, hence we possibly  attain the best generalization score that can be achieved using these architectures and datasets. We do believe that these methods can be further improved, allowing for larger architectures and bigger datasets, and as such better generalization.
>
>
> 2) This is an interesting question. We compare our results to baseline models that are trained without regularization at all, for example see the baselines in Fig.6 (as dashed lines) and the rightmost column in Fig.4. The reason we specifically used weight decay as a regularizer was because of the theoretical understanding we have about it (presented in Section 5). Studying memorization in models trained with other regularizers is an interesting direction and we currently don't have a theoretical understanding of how it would affect reconstructability. Testing this question empirically may prove to be useful, but is beyond the scope of this paper.
>
> 3) We agree that the datasets we trained on are relatively small scale. We emphasize that this is also a limitation of previous works on data reconstruction. Note however that in our work we were able to reconstruct from a model trained on $5$,$000$ samples, which is $5$ times more than the largest dataset in [Haim et al. 2022]. Reconstructing from ImageNet is a challenging and non-trivial task (even when considering only a small part of it). We believe it is possible, but may require further improvement of the current reconstruction methods and theoretical understanding, and we leave it for future research.
>
> Regarding limitations, similar to previous works on data reconstruction our results are limitied to relatively small datasets and architectures. We will emphasize these in the camera-ready version by adding a limitations section.

---

> > ### Comment · Reviewer_FQAU · 2023-08-21
> >
> > Thank the authors for taking the time in answering my questions. After reading the rebuttals and the reviews from other reviewers and the discussions, some of my previous concerns of this paper remain unaddressed, therefore, I would like to stick to my original evaluation.

---

### Official Review · Reviewer_Poqu · 2023-07-05

**Soundness:** 3 good
**Presentation:** 3 good
**Contribution:** 3 good
**Rating:** 7
**Confidence:** 4

**Summary:**

This paper is following the work of Haim et al. (2022) who introduced a reconstruction scheme for neural network with logistic or exponential loss for binary classification tasks. They extended this work by introducing this reconstruction scheme in a multi class setting. They also demonstrated how it can also be extended to convolutional neural network and more general loss functions.

**Strengths:**

- This paper is well written and the problematic well presented.
- Memorization is indeed an important research area and this paper offer significant finding that give a better understanding on which conditions influence memorization of training data.
- I really appreciated the balance between quantitative and qualitative experiment. As well as the ablation study with the number of training class and examples. It gives really good insights.
- The appendix contain good explanations and interesting additional experiments.

Overall, I think this is a good paper that give important insights for the community.

**Weaknesses:**

- One small issue I have with this paper is that there isn't a single strong storyline. It is kind of a mix of different improvements over Haim et al. (2022) with General Loss Function/Multi Class and CNN. So it's like section 4, 5 and 6 could be entirely independent. For example, I was surprised after introducing the multi class that the CNN was trained with a binary classification task. In term of presentation I would probably have started with the General loss, then present the multi class setting, then present multi-class with CNN (and weight decay) (Just to make the story smoother).

**Questions:**

Is there any reason why you didn't use a multi class setting with the CNN ?

**Limitations:**

I didn't found a limitation section in this work or in the appendix. One main limitation is that this work is limited to extremely small architectures and that it's not clear how much of it will transfer to practical architectures.

---

> ### Author Rebuttal · Authors · 2023-08-09
>
> We thank the reviewer for the positive and constructive feedback.
>
> Regarding the paper's storyline: this is a good point, and we had a dilemma on how to organize the different sections. We will consider implementing the reviewer's suggestion for reordering in the camera-ready version.
>
> Regarding multi-class CNNs: the reason is that we wanted to decouple the effects of changing the architecture from changing to multiclass from binary classificatoin. We can add a multiclass CNN experiment in the camera-ready version.
>
> Regarding limitations, we will add a limitations section in the camera-ready version with an elaborated discussion on the limitations of our current approach, including references to small architectures and datasets.

---

> > ### Comment · Reviewer_Poqu · 2023-08-14
> >
> > Thank you for your answer. I looked at the other reviews, answers and I will keep my score. The main weakness that is raised by another review is about "novelty". However, novelty is always a very subjective notion and researchers often have different threshold of what they consider something novel. Since the authors are transparent of the fact that their work extends [Haim et al. 2022] and that they provide additional insights (with a rigorous experimental setup) which were not demonstrated before, I consider that this paper should be accepted.

---

### Official Review · Reviewer_ZiTy · 2023-07-10

**Soundness:** 3 good
**Presentation:** 3 good
**Contribution:** 2 fair
**Rating:** 4
**Confidence:** 4

**Summary:**

This work extends the work of [Haim et al. 2022] to multi-class classification and with a more general loss function. Through mainly empirical evaluations, the authors demonstrate that data reconstruction is also possible and reveal interesting relationships between the ability to reconstruct and the number of classes, number of training data, size of the network, and the role of weight decay in the process.

The findings are most empirical. Though interesting, the novelty and technical contribution of this work is relatively low.

**Strengths:**

Reconstructing samples from trained classifiers is an important problem with direct connections to generative modeling and model privacy. The findings of this paper, though derivative from [Haim et al. 2022], are interesting and non-trivial. The findings regarding the data reconstruction ability vs. number of classes and weight decay are novel to me. They do bring new insights into understanding neural network classifiers.

**Weaknesses:**

Novelty: This is my main concern about this submission. The framework is a direct extension from [Haim et al. 2022]. The objectives have been slightly modified, for the multi-class cross-entropy case and for the general loss with weight decay case, but they do not bring significant new insights over the original work.

Experiments: Though the experiments conducted in this work are interesting, the results do not include variation analysis or error bars from multiple runs, nor statistical tests for whether two cases are significantly different. The experiments can be further improved by
1) evaluating large-scale datasets, e.g., ImageNet, rather than the CIFAR10 dataset, which has been investigated in [Haim et al. 2022].
2) extending from simple classifier architectures (MLPs, Simple CNNs) to more complicated architectures such as ResNet or Vision Transformers.

Analysis: The empirical findings lack depth. For instance, the observations about weight decay vs. data reconstruction seem new and interesting. Is any of them provable, even in oversimplified cases? There are theoretical works on classifiers trained with square loss and weight decay [1].  Adding more theoretical justifications will make this submission stronger.
I feel that this is a missed opportunity and I will consider raising my score if more theoretical insights can be provided to support the empirical findings.

[1] Hu et al. Understanding square loss in training overparametrized neural network classifiers

**Questions:**

For empirical evaluations, I am wondering how good are the reconstructed samples in terms of FID. There is a line of work trying to generate/reconstruct samples from the classifier, mostly by gradient ascent [2,3]. How does your method compare with those as a generative model? Are there any deeper connections between the two very different methodologies?

[2] Wang and Torr, Traditional Classification Neural Networks are Good Generators: They are Competitive with DDPMs and GANs

[3] Zhu et al. Towards Understanding the Generative Capability of Adversarially Robust Classifiers

**Limitations:**

The authors have addressed the potential negative societal impact. The limitation of the work could have been more thoroughly discussed.

---

> ### Author Rebuttal · Authors · 2023-08-08
>
> We thank the reviewer for the thorough review and the constructive comments.
>
> 1) Regarding novelty, we would like to emphasize that although our work is a direct extension of [Haim et al. 2022], we make some significant and novel contributions and overcome some of the limitations of the previous work. To name a few:
>
> - We generalized the previous work to a multi-class setting (as opposed to binary classification). This required a non-trivial extension of the KKT solution to the max-margin problem, which was not shown before in practical applications.
>
> - We used a different theoretical justification (beyond convergence to a KKT of the max margin problem) to devise a reconstruction loss which works on general regression losses, and demonstrated it on three different losses.
>
> - We investigated the connection between weight decay, number of parameters and reconstructability and found a connection which was not known before. This connection enabled us to overcome some of the limitations of the previous work, such as reconstruction from CNNs (as opposed to reconstruction from MLPs only), reconstruction from models with a standard initialization scheme (as opposed to models trained with a very small and non-standard initialization scale), and reconstruction from a model trained on up to $5$,$000$ samples. As far as we know, this is the largest dataset used to train a classifier and shown to be successfully reconstructed.
>
> 2) Regarding the experiment and improving to larger datasets and more complicated architectures: we believe that the end goal of understanding memorization in neural networks is to be able to apply it on more realistic settings. However, the current state of the research in this field is still in its infancy, and we believe that our work provides an important stepping stone in that direction. We agree that there is much more work to be done for future research in this field.
> Regarding the comment about the error bars,  we will add them whenever relevant in the camera ready version. We note that every reconstruction experiment in our paper uses a large hyper-parameter sweep and over many different runs. The full details appear in Appendix B.
>
> 3) Regarding theoretical justifications, we thank the reviewer for the helpful comment. Please note that of the three main sections in this work, two provide theoretical analysis and insights on the success of the demonstrated empirical reconstruction results (Sections 4\&5).
>
>     We also have a theoretical justification on why weight decay helps reconstructability for simplified networks, which we will add in the camera-ready version.
>
>     In a nutshell, the (directional) convergence rate to a KKT point of the max-margin problem, as paraphrased in Thm 3.1, might be extremely slow -- logarithmic in $t$ for  time $t \to \infty$. Hence, even after many gradient descent steps (e.g., $10^6$ iterations) the learning process may essentially stop because of small gradients, but the parameters may still be far from the direction of a KKT point. A relation between small initialization and such convergence is shown in Moroshko et al. [1]. They prove that for diagonal linear networks (i.e., a certain simplified architecture of deep linear networks) the initialization scale controls the rate of convergence to the KKT point. Namely, when the initialization is small, gradient flow converges much faster to a KKT point.
>
>     Simply put: without weight-decay, small gradients (of the empirical loss) do *not* imply convergence to a KKT point. However: small initialization implies faster convergence (to KKT), which implies better reconstructability.
>
>     In our novel approach - when weight-decay is used, small gradients (of the loss) *do* imply better convergence to the solution in Eq. 14, as opposed to convergence to a KKT point in the case of training without weight-decay. And better convergence implies better reconstructability. This provides a theoretical justification for why incorporating weight decay may result in better reconstructability for models with standard initialization, which is one of the main contributions in this work.
>
> 4) Regarding evaluation with FID, this is an interesting question and an interesting comparison that could be made. However, we emphasize that our work is focused on understanding *memorization* in neural networks, while works on generative models focus on how to model a distribution. Hence, such comparisons are out of the scope of our paper.
>
> [1] Edward Moroshko, Blake E. Woodworth, Suriya Gunasekar, Jason D. Lee, Nati Srebro, and Daniel Soudry. "Implicit bias in deep linear classification: Initialization scale vs training accuracy." Advances in neural information processing systems 33 (2020): 22182-22193.

---

> > ### Comment · Reviewer_ZiTy · 2023-08-13
> >
> > Thanks for the rebuttal! Most of my raised questions have been addressed.
> >
> > However, my biggest concern still stands, i.e., the novelty of this work on top of existing literature.
> >
> > I thank the authors for listing 3 bullet points for extra contribution. Could the authors elaborate on which one is the most significant, and the associated technical difficulties?
> >
> > In my humble opinion, extending the methodologies from binary classification to multi-class (equation 12) is relatively straightforward.
> > About the general losses case, equation 15 also seems a bit derivative from equation (6). The implicit bias of minimum $l_2$ norm is substituted by an explicit penalty on $l_2$ norm, which also doesn't seem surprising in terms of extending the methodology.
> >
> > I could be missing some important points here and hopefully the authors could explain more.

---

> > > ### Author Response · Authors · 2023-08-14
> > > **Re: Comment**
> > >
> > > We thank the reviewer for the comment.
> > >
> > >
> > > Regarding the extension to general regression losses (Eq. (15)), the novelty in this formulation is that it is *conceptually* different than reconstruction from classification. Note that in [1], the theoretical framework, which is the main basis for that work, revolved only around classification losses.
> > > Their reconstruction method is based on the KKT conditions for margin maximization, which allows for reconstructing the points on the margin.
> > > Showing reconstruction in regression setup is not a trivial extension of [1], both conceptually, and also it is not trivial to assume that the success of the reconstruction in the classification case would apply in the regression case - empirically speaking.
> > >
> > > Moreover, the results that we show in the paper are not only empirical. We also based our results on the theoretical reasoning (discussed in Section 5).
> > > Although the loss in Eq. (15) looks similar to the one in Eq. (6), the theoretical reasoning behind the two losses is very different. They are also not exactly the same. Note that in Eq. (15) there is no restriction on the $\lambda_i$'s, namely, they may be negative. As opposed to Eq.(6) where $\lambda_i$ are required to be positive.
> > >
> > >
> > > We also emphasize that the role of weight decay and the effect of the number of parameters on reconstructability was not known before.
> > > One of the main limitations of [1] is that their method required very small and not-so-commonly used initialization scheme. We overcome this limitation by introducing weight decay, which is a common practice. Another limitation of [1] is that they focused on fully-connected networks, while adding weight decay allowed us to extend the reconstruction to CNNs.
> > >
> > >
> > >
> > > Regarding reconstruction from a multiclass classifier: the reconstruction loss in Eq.(12) is not the trivial extension of the max-margin conditions for multi-class classification.
> > > A straightforward extension of the KKT conditions for the multiclass case, as appears in Appendix G in [2], did not work as-is for reconstruction. Only after formalizing the equivalent form, by removing most of the constraints (as discussed in lines 137-150), did we manage to get good reconstructions from multiclass classifiers.
> > >
> > >
> > > We agree that assessing the novelty of works in general is rather subjective. However, we believe that our work extends previous works on reconstruction in several directions, far beyond what was known before.
> > >
> > >
> > >
> > >
> > > [1] Reconstructing Training Data from Multiclass Neural Networks, Niv Haim, Gal Vardi, Gilad Yehudai, Ohad Shamir, Michal Irani, 2022
> > >
> > > [2] Gradient Descent Maximizes the Margin of Homogeneous Neural Networks, Kaifeng Lyu, Jian Li, 2019

---

> > > > ### Comment · Reviewer_ZiTy · 2023-08-17
> > > >
> > > > I thank the authors for the elaboration. The relationship to previous works and the unique contributions of this work are more clear to me. However, I still have concerns about the novelty and the depth of the empirical findings. I am raising my score to 4 to reflect my current assessment of the work.
> > > >
> > > > About the question I raised in my initial review, about related work [2] and [3]. The response is not satisfactory. The authors wrote
> > > > "However, we emphasize that our work is focused on understanding memorization in neural networks, while works on generative models focus on how to model a distribution."
> > > >
> > > > Although [2] and [3] report the generative capability of trained classifiers, I think they can also be seen as studying memorization in neural networks. They are not separately trained generative models, but pre-trained classifiers.
> > > >
> > > > The perspective may be different though.
> > > > [2] and [3], and lots of other works, study **recovering the whole training set**. The evaluation metric is FID. Your work focuses on **individual sample recovery**. That is why I am asking the question. I am wondering what the advantage of your method is compared to the more intuitive gradient ascent, which is widely applicable for general losses and structures.
> > > >
> > > > --------------------------------------------------------------
> > > >
> > > > I have another question regarding why in general, recovering training sample from trained neural networks is possible (this does not affect my review).
> > > > Consider a simple case where we have a 1D Gaussian mixture to classify.
> > > > Let $P(x|y=1) \sim N(\mu, \sigma^2)$ and $P(x|y=-1) \sim N(-\mu, \sigma^2)$. The Bayes optimal classifier is $Sigmoid(2\mu x/\sigma)$, where $\mu$ and $\sigma$ is confounded. Can we recover information for $\mu$ alone given a pre-trained classifier?
> > > >
> > > > In the above case, gradient ascent type methods, e.g., [2], tend not to work and the optimization will diverge to infinity. Does your method have advantages in this case?
> > > >
> > > > I feel this type of discussion is helpful for the readers about the topic. Not only neural networks, but also the training data itself, plays a critical role in data reconstruction.

---

> > > > > ### Author Response · Authors · 2023-08-19
> > > > >
> > > > > We thank the reviewer for their response and for raising their score.
> > > > >
> > > > > Regarding [2] and [3], we agree that both works shed some light on memorization, as all other works on visualization, since both topics are related.
> > > > > However, they do not directly deal with how specific training samples are memorized or how they can be reconstructed from a classifier.
> > > > >
> > > > > The way we see it:
> > > > >
> > > > > - [2,3] show recovery of images from pretrained classifiers, similar to other works on visualization that we discuss in the "related works" section. Please note that there is no evidence in [2,3] of recovering even a single example from the training set. We emphasize that none of these previous works show recovery of the "whole training set".
> > > > >
> > > > > Regarding FID:
> > > > >
> > > > > - There is not much point in using FID in our work. For instance, many reconstructed candidates do not resemble training samples at all and will dramatically increase FID score, or any other metric for comparing distributions. But this makes sense, since we do not aim at studying generative models or modeling the distribution of the training set.
> > > > > (BTW, note that FID is also not reported in [2], only a somewhat suspicious "differentiable FID" whose nature is not explained in the paper).
> > > > >
> > > > > - [3] does not deal with standard trained classifiers. It deals with classifiers that are either adversarially trained, or whose training process is altered in manners like injecting high noise that would make them good "generators". So in this case it makes sense to evaluate with FID.
> > > > >
> > > > > Regarding your question on the 1D Gaussian mixture: Even in this simple setting, we believe that it is non-trivial to show a theoretical guarantee regarding information recovery with our method. We agree that analyzing information recovery in this setting, both theoretically and empirically, is an intriguing question.

---

> > > > > > ### Comment · Reviewer_ZiTy · 2023-08-20
> > > > > >
> > > > > > "We emphasize that none of these previous works show recovery of the "whole training set""
> > > > > >
> > > > > > I couldn't agree with this argument. The FID is usually calculated between the generated samples and the training data, for evaluating generative models. When using FID to measure reconstructed images from trained classifiers, e.g., [2,3], I think it does reflect recovery of the "whole training set", in the sense of matching distribution. It is true that they do not focus on each specific sample, but I think the goal is shared with this submission. IMHO, if [2,3] had a **lookup procedure to find the closest training image for each generated image, they can also directly compare with your work**. Maybe your method is way better in sample-level reconstruction. But the comparison is not made.
> > > > > >
> > > > > > I admire the work of [Haim et al. 2022] since they offered a very different and theory-inspired method for reconstructing data from trained classifiers.
> > > > > > This work generalized the methodology of [Haim et al. 2022] to multi-class and general losses, where **gradient-ascent-like methods like [2,3] are already doing similar things**.  I am wondering whether your method is better for dataset reconstruction or model inversion. **I think such comparisons are necessary and important for the readers of this work**.

---

> > > > > > > ### Author Response · Authors · 2023-08-20
> > > > > > >
> > > > > > > First, in Figure 7 of [3] they actually compare their generated images to nearest images from the dataset (in terms of SSIM) in order to show that their generated images are not similar to training examples.
> > > > > > >
> > > > > > > Second, in Haim et al. they showed that gradient ascent is not able to reconstruct training examples with their models (see Section 5.4, Figure 5 and Appendix C in their work). We thought that repeating the same experiments as Haim et al. does not add value to our work.

---

### Decision · Program_Chairs · 2023-09-21

**Decision:**

Accept (poster)

**Comment:**

Building on previous work regarding training data reconstruction from MLPs trained to perform binary classification, this submission describes a strategy to reconstruct training samples from both MLPs and CNNs trained with a larger family of loss functions. It further analyzes the effect of weight decay and model width upon the efficacy of the described method.

Reviewers agreed that the work tackles an important problem and contains novel elements, although the method appears to be limited to shallow networks trained on small datasets. One reviewer raised concerns regarding the novelty and the magnitude of the contribution relative to Haim et al. (2022), upon which it builds. A second reviewer raised concerns regarding the absence of a strong storyline, but nonetheless supported acceptance.

I feel that the methodology and theory described in this submission are sufficiently novel. The method, while limited in its current form, could lead to practically meaningful data reconstruction methods and/or techniques to limit memorization. Thus, I recommend acceptance.